# Decoding the IL-33/ST2 Axis: Its Impact on the Immune Landscape of Breast Cancer

**DOI:** 10.3390/ijms241814026

**Published:** 2023-09-13

**Authors:** Bojan Stojanovic, Nevena Gajovic, Milena Jurisevic, Milica Dimitrijevic Stojanovic, Marina Jovanovic, Ivan Jovanovic, Bojana S. Stojanovic, Bojan Milosevic

**Affiliations:** 1Department of Surgery, Faculty of Medical Sciences, University of Kragujevac, 34000 Kragujevac, Serbia; bojan.stojanovic01@gmail.com (B.S.);; 2Center for Molecular Medicine and Stem Cell Research, Faculty of Medical Sciences, University of Kragujevac, 34000 Kragujevac, Serbiaivanjovanovic77@gmail.com (I.J.); 3Department of Pharmacy, Faculty of Medical Sciences, University of Kragujevac, 34000 Kragujevac, Serbia; milena.jurisevic13@gmail.com; 4Department of Pathology, Faculty of Medical Sciences, University of Kragujevac, 34000 Kragujevac, Serbia; 5Department of Otorinolaringology, Faculty of Medical Sciences, University of Kragujevac, 34000 Kragujevac, Serbia; 6Department of Pathophysiology, Faculty of Medical Sciences, University of Kragujevac, 34000 Kragujevac, Serbia

**Keywords:** Interleukin-33, breast cancer, tumor microenvironment, immunotherapy, metastasis

## Abstract

Interleukin-33 (IL-33) has emerged as a critical cytokine in the regulation of the immune system, showing a pivotal role in the pathogenesis of various diseases including cancer. This review emphasizes the role of the IL-33/ST2 axis in breast cancer biology, its contribution to cancer progression and metastasis, its influence on the tumor microenvironment and cancer metabolism, and its potential as a therapeutic target. The IL-33/ST2 axis has been shown to have extensive pro-tumorigenic features in breast cancer, starting from tumor tissue proliferation and differentiation to modulating both cancer cells and anti-tumor immune response. It has also been linked to the resistance of cancer cells to conventional therapeutics. However, the role of IL-33 in cancer therapy remains controversial due to the conflicting effects of IL-33 in tumorigenesis and anti-tumor response. The possibility of targeting the IL-33/ST2 axis in tumor immunotherapy, or as an adjuvant in immune checkpoint blockade therapy, is discussed.

## 1. Introduction

Breast cancer, characterized by an uncontrolled proliferation of mammary gland cells, is the most frequently diagnosed malignancy in women globally [1]. With an estimated 2.2 million cases and approximately 700,000 fatalities in 2021, there is an indisputable urgency to advance early detection techniques, innovate therapeutic interventions, and extend the realm of research [1]. These alarming statistics underscore the profound need for progress towards the effective management and, ultimately, the eradication of this substantial global health burden.

Comprising approximately 14% of all cancer diagnoses, breast cancer remains the second most common malignancy worldwide [2]. For women aged 20 to 59, it is the primary cause of cancer-related mortality, highlighting the crucial necessity for enhanced early detection methodologies and efficacious treatment regimens [3]. Importantly, the principal determinant of mortality in breast cancer is not the primary tumor, but its metastatic dissemination to distant organs, making the elucidation of the underlying mechanisms of this metastatic progression critical for improved clinical outcomes and comprehensive disease management [4].

Breast cancer’s complexity, as the predominant malignancy among women, stems from its striking heterogeneity and multifaceted clinical manifestations [5]. The variation in phenotypes and morphological types underscores the multidimensional nature of this disease, influencing its characterization and treatment strategies [6]. The disease’s intricacies extend to the classifications of Hormone Receptor (HR)-positive (estrogen and progesterone hormone receptors), Human Epidermal Growth Factor Receptor 2 (HER2)-positive, and Triple-Negative Breast Cancer (TNBC), each presenting unique challenges and insights in the management of breast cancer [7]. Such heterogeneity necessitates personalized therapeutic strategies, introducing an added layer of complexity to this multifaceted disease [8].

Amid this intricate landscape, Interleukin-33 (IL-33), a member of the IL-1 cytokine family, has gained recognition as a potentially significant molecule in this context [9]. IL-33, typically secreted in response to cellular damage, functions as an alarmin [10]. Our previous observations of positive correlations between systemic IL-33, IL-6, and IL-12 serum levels in patients with COVID-19 hint at the synergistic effects of these cytokines in disease pathogenesis [11]. However, in oncogenesis, IL-33 has been shown to exert dual roles, both inhibitory and stimulatory, on tumor tissue, with the balance likely dependent on the tumor’s immunogenicity, microenvironment, and IL-33 localization [12].

This review aims to dissect the emerging role of IL-33 within the framework of breast cancer, seeking to illuminate its potential implications on detection, prognosis, and treatment paradigms.

## 2. Unraveling the Enigma of Interleukin-33 in the Cancer Landscape

Interleukin 33, an integral cytokine of the IL-1 family, has emerged as a key player in the pathophysiology of various inflammatory and allergic diseases, as well as certain cancers [13]. Yet, its exact role remains enigmatic. Identified as a potent stimulator of immune responses and a contributor to wound healing, the complex role of IL-33 in modulating the Tumor Microenvironment (TME) is increasingly attracting attention [14,15]. Its ability to orchestrate immune cell recruitment within the TME may influence tumor phenotype and malignancy, making it a potential target of interest in cancer research [16].

Since its discovery in 2005, IL-33 has been recognized as a vital cytokine in the IL-1 family, celebrated initially for its capacity to ignite type 2 immunity [17]. Moreover, recent findings have propelled IL-33 to the forefront as a pivotal immune regulator boasting diverse roles in both type 2 and type 1 immune responses, as well as regulatory activities. Beyond its well-established effects on type 2 immune cells such as Innate Lymphoid Cells type 2 (ILC2s), mast cells, T-helper 2 (Th2) cells, eosinophils, basophils, and dendritic cells, research over the past years has unveiled IL-33’s influential capacity to also modulate cells central to type-1 immunity and chronic inflammation. This includes cells like T-helper 1 (Th1) cells, Natural Killer (NK) cells, CD8+ T cells, neutrophils, macrophages, B cells, and Natural Killer T (NKT) cells, showcasing the cytokine’s multifaceted impact on the immune landscape [13].

The IL-33/ST2 axis plays a crucial role in myriad conditions, extending beyond allergies to span obesity, Alzheimer’s disease, and cancer [18]. Intriguingly, IL-33 has earned the title of an “alarmin” or damage-associated molecular pattern (DAMP) due to its capacity to bolster type 1 antiviral immunity when dispatched from virally infected epithelial barrier cells [13].

While many propose that IL-33 is pro-tumoral in humans, an accumulating body of evidence from preclinical mouse models suggests the contrary [19]. External IL-33 may stimulate a robust type 1 antitumor immune response [20]. Interestingly, several human studies indicate that endogenous IL-33 might function as an antitumor agent, with its absence in tumor tissue potentially fostering metastasis [21].

### 2.1. Comprehensive Understanding of Interleukin-33: Structure, Function, and Regulation

Interleukin 33 is a versatile protein that functions as both a transcription factor and a cytokine, playing a vital role in inflammatory responses [9]. IL-33 is ubiquitously found across a broad array of tissues and cells, and its expression is heightened by various stimuli, primarily functioning as a pro-inflammatory cytokine [22].

The unique structure of IL-33 is characterized by dual features: an N-terminal region equipped with a nuclear localization signal and a DNA-binding domain, and a C-terminal region resembling an IL-1 cytokine domain (Figure 1) [23]. This dual nature enables IL-33 to function both intracellularly and extracellularly [24]. Intracellularly, IL-33 targets the nucleus, binding to chromatin, and potentially modulating gene expression through chromatin remodeling and epigenetic mechanisms [25]. Conversely, in the event of cellular damage or infection, IL-33 transitions from a nuclear protein to a secreted cytokine, acting as an “alarmin,” signaling cellular distress [24].

A unique feature of IL-33 is its regulation, differing from the typical behavior of IL-1 family members [26]. Its regulation depends on a careful balance of activation and inactivation via different caspases and proteases. Importantly, it does not require caspase-1 cleavage to function as a cytokine. Instead, full-length IL-33 is processed by apoptosis-associated caspases 3 and 7, which act as a shutdown mechanism, curbing IL-33’s pro-inflammatory activity and preventing unwanted secretion [27,28]. On the other hand, under inflammatory conditions, human and mouse full-length IL-33 are cleaved by serine proteases, producing highly potent mature forms of IL-33 [29]. Mast cells and neutrophils, significant in these processes, are actively recruited into the tumor microenvironment [29,30]. The mature forms of IL-33 behave like IL-1-like cytokines, amplifying the potency of IL-33 within an inflammatory environment and highlighting the complex regulation and dynamic nature of IL-33 [31].

IL-33 is primarily detected in non-immune cells such as epithelial barrier cells and fibroblasts and triggers intracellular molecular pathways through its receptor ST2 (IL-1R-like-1) [32]. This receptor is found across a wide spectrum of both innate and adaptive immune compartments, influencing cells like ILC2s, mast cells, eosinophils, natural killer cells, dendritic cells, and T cells [33].

The IL-33 locus is found on chromosome 9 at 9p24.1 in humans and 19qC1 in mice [26]. Several splice variants of IL-33 have been identified, which affect protein activity [34]. The IL-33 protein incorporates a nuclear/chromatin-binding region, a cleavage region, and a consensus region for caspase cleavage [33]. Its interaction with nucleosomes confines its expression to the nucleus [35]. The IL-33 gene in humans spans approximately 42 kb within genomic DNA and encompasses eight exons [13]. The resulting protein consists of 270 amino acid residues, arranged in a β-trefoil fold resembling other family members [26]. IL-33 may function as an endogenous danger signal, released in response to viral threats and during necrosis [22]. Infection and autoimmune diseases often prompt IL-33’s release, further illustrating the critical role of this cytokine in immune responses [24]. The IL-33 molecule is comprised of three recognized functional domains: a nuclear domain, a central domain, and an IL-1-like cytokine domain [32]. A homeodomain-like helix-turn-helix configuration in the N-terminus is essential for its nuclear relocation, indicating the intricacy of its structural design [23].

The context-dependent regulation and function of IL-33 is another intriguing aspect of its biology [26]. Under conditions of stress or infection, the full-length IL-33 can be released from the cells, functioning as an alarmin or danger signal, enhancing inflammation when released from necrotic cells [22]. However, during apoptosis, IL-33 is neutralized by caspase-3 or caspase-7, which serves as a safety mechanism to prevent unwanted inflammation under non-pathological conditions [28]. Apart from the cellular distress signal, IL-33 also exhibits increased activity under inflammatory conditions [24]. For instance, full-length IL-33 can be cleaved by the serine proteases cathepsin G and elastase, yielding more potent mature forms of IL-33 [29]. This capacity to upregulate its bioactivity in an inflammatory microenvironment underscores its significant role in disease processes, and further studies are warranted to elucidate its specific role in health and disease.

In summary, IL-33 is a multifaceted cytokine playing a vital role in inflammatory responses and immune regulation. It is tightly regulated at both intracellular and extracellular levels and modulates the immune response depending on the specific environmental context. With its distinct structure, diverse roles, and complex regulation, IL-33 continues to be a subject of extensive research in understanding disease processes and developing potential therapeutic strategies.

### 2.2. Characteristics and Functions of the IL-33 Receptor (ST2) Isoforms

The Suppression of Tumorigenicity 2 (ST2) receptor, initially identified in murine fibroblasts as an oncogene-induced gene, predates the discovery of its ligand, IL-33 [21]. It is encoded by the IL1RL1 gene and exists in four alternative splicing isoforms: ST2L (ligand), soluble ST2 (sST2), ST2V (variant), and ST2LV (ligand variant) [36].

ST2L is a membrane-bound variant with structural features reminiscent of IL-1 type-1 receptors [37]. It consists of three Ig-like extracellular domains, a transmembrane region, and an IL1-R1-like intracellular domain [38]. This isoform collaborates with Interleukin-1 Receptor Accessory Protein (IL1-RAcP) to form a heterodimeric transmembrane receptor complex that initiates signal transduction upon IL-33 binding [38]. ST2L is expressed by a range of cells, including fibroblasts, mast cells, eosinophils, Th2 lymphocytes, dendritic cells, basophils, invariant natural killer cells (iNKT), and macrophages [36]. Conversely, sST2 is a soluble form of ST2 that, although lacking the transmembrane domain, retains an extracellular domain similar to ST2L. Acting as a decoy receptor, sST2 sequesters extracellular IL-33, thereby modulating its activity. Fibroblasts and epithelial cells primarily express this isoform [36].

The other two isoforms, ST2V and ST2LV, are less well-studied. ST2V has a hydrophobic tail in place of the third Ig-like domain, whereas ST2LV is a soluble, N-glycosylated variant devoid of ST2L’s transmembrane domain [21].

The expression of ST2 isoforms is finely tuned by GATA1/2 and estrogen-response elements (EREs). Interestingly, both isoforms, ST2L and sST2, can be transcribed from either distal or proximal promoters, with the cell type appearing to dictate the preferred promoter [21].

### 2.3. The Intricate Dynamics of IL-33/ST2 Signaling

IL-33, a dual-function molecule, plays a critical role in orchestrating inflammatory responses [39]. As a nuclear factor, IL-33 interacts with chromatin to suppress inflammation [40]. However, when extracellularly released due to cell damage or mechanical injury, it operates as a cytokine and binds to the membrane-bound ST2L receptor, prompting a series of molecular events [31].

Upon binding, IL-33 triggers a conformational change in ST2L, instigating the assembly of a heterodimeric receptor complex with IL-1RAcP on the cell membrane [38]. This assemblage recruits adaptor molecules including Myeloid Differentiation Primary Response 88 (MyD88), Interleukin-1 Receptor-Associated Kinase 1 (IRAK-1), IRAK-4, and TNF Receptor Associated Factor 6 (TRAF6), initiating the transduction of the IL-33 signal [41]. The subsequent degradation of the Inhibitor of kappa B (IκB) protein results in the activation of the Nuclear Factor kappa-light-chain-enhancer of activated B cells (NF-κB) transcription factor, while the Mitigen-Activator Protein (MAP) kinases (p38), c-Jun N-terminal kinases (JNK), and Extracellular signal-regulated kinases (ERK) stimulate transcription factors like Activator Protein 1 (AP-1), ultimately driving the expression of specific cytokines and chemokines [21].

The IL-33/ST2 pathway plays a significant role in Th2 responses, stimulating the production of cytokines like IL-4, IL-5, and IL-13 in Th2 cells, and fostering chemokine production in epithelial cells [42,43]. Additionally, in combination with the pro-inflammatory cytokine IL-12, IL-33 enhances the production of Th1 cytokine IFN-γ [44]. In the context of cancer, the IL-33/ST2 signaling pathway influences the tumor microenvironment by activating immune effector cells and regulating the recruitment of pro- or anti-tumorigenic cells [19].

However, the IL-33/ST2 signaling is balanced through various regulatory mechanisms [36]. IL-33’s pro-inflammatory activities are modulated through its nuclear localization, its degradation by pro-apoptotic proteases, and the attenuation of its signaling under stressful conditions [45]. The ST2L receptor itself undergoes internalization and proteasome-targeted degradation when phosphorylated by enzymes like Glycogen Synthase Kinase 3 beta (GSK-3β). Additional disruption occurs through the Single Immunoglobulin IL-1-Related Receptor (SIGIRR), which interrupts the ST2/IL1-RAcP heterodimer, and the soluble form sST2, which sequesters IL-33, precluding its interaction with ST2L. Moreover, IL-33 can be rapidly inactivated extracellularly by the oxidation of its cysteine residues and disulfide bond formation [21].

The formation of the ST2, IL-1RAcP, and MyD88 complex enables the recruitment of IRAK1,4 and TRAF6, activating key signaling pathways such as the Janus Kinase-Signal Transducer and Activator of Transcription (JAK-STAT), NF-κB, JNK, ERK, and p38. These intricate relationships and mechanisms, revealed through the examination of the crystal structures of IL-33 and ST2, and the IL-33/ST2 complex, underscore the essential role IL-33/ST2 signaling plays in immune responses [21]. The signaling cascade following the binding of IL-33 to ST2, as well as the outcomes of this signaling, are depicted in Figure 2. This visualization further elucidates the complex interplay of molecules involved in this critical immunomodulatory pathway, reinforcing the importance of IL-33/ST2 in regulating immune responses.

## 3. IL-33 and Its Implications in Human Breast Cancer

The complex role of IL-33 in the progression of breast cancer has gained significant attention, underscored by numerous patient-based studies [21]. In particular, it has been observed that individuals diagnosed with breast cancer typically present with considerably elevated serum levels of IL-33, suggesting a potential role for this cytokine as a biomarker for this malignancy [16,46]. Several recent studies have ventured into understanding the implications of IL-33 and its soluble receptor, sST2, in the context of breast cancer [47]. These investigations offer valuable insights that could drive advancements in both diagnostic and therapeutic strategies for this disease. Elevated concentrations of IL-33 and sST2 have been identified in the serum of breast cancer patients in comparison to healthy individuals [47]. This significant disparity in serum levels underscores the potential of IL-33 and sST2 as biochemical markers, potentially enhancing early detection strategies. Nevertheless, it is worth noting that some studies have reported no observable differences in serum concentrations, highlighting the necessity for further research to validate these findings across diverse populations and settings [46]. 

Interestingly, serum concentrations of IL-33 and sST2 show a strong correlation with Vascular Endothelial Growth Factor (VEGF), a key player in angiogenesis, the process by which new blood vessels form. This association may imply that IL-33 and sST2 could have a role in fostering angiogenesis, thereby facilitating tumor growth and progression [47,48].

Furthermore, notable associations have been observed between IL-33 and sST2 levels and the concentrations of Matrix Metallopeptidase 11 (MMP-11) and Platelet-Derived Growth Factor-C (PDGF-C). These factors play essential roles in tissue remodeling and cell proliferation, thereby further implicating IL-33 and sST2 in the complex dynamics of breast cancer progression [47].

A comparison between benign breast cancer patients and those with carcinoma has revealed higher serum IL-33 levels and tissue expression in the latter group [46]. Moreover, increases in serum IL-33 levels appear to be more pronounced in advanced stage IV breast cancer, suggesting a role for IL-33 in disease severity and progression [21].

However, investigations into IL-33 genetic polymorphisms and serum levels among breast cancer patients and healthy individuals have found no significant differences in the frequencies of genotypes and alleles at specific single nucleotide polymorphisms (SNPs) in the IL-33 gene. Moreover, these genotypes do not appear to influence the serum levels of IL-33 in either patients or healthy controls [49].

Another intriguing aspect of IL-33 involves its potential role in the differential diagnosis of idiopathic granulomatous mastitis (IGM) and breast cancer. Serum IL-33 levels have been found to be higher in patients with IGM compared to those with breast cancer, suggesting IL-33 and sST2 could serve as potential differential diagnostic markers in conjunction with radiological and pathological examinations [50].

In summary, IL-33 and its receptor sST2 hold promising potential as key players in the complex landscape of breast cancer progression and diagnosis. Further investigations are warranted to confirm these findings and explore their implications for therapeutic interventions.

### 3.1. The Influence of IL-33 on ER-Positive Breast Cancer

The role of IL-33 and its receptor, soluble ST2, in estrogen receptor (ER)-positive breast cancer progression has been highlighted in extensive patient-based research [51]. These studies show that elevated serum levels of both IL-33 and sST2 are not just characteristic of ER-positive breast cancer but also correlate with the concurrent increase in VEGF and other angiogenic factors such as MMP-11 and PDGF-C [47]. This compelling evidence suggests that IL-33 and sST2 serve as key markers for unfavorable prognosis and decreased survival rates in ER-positive breast cancer patients. Additionally, the data highlight the potential role of sST2 as a crucial factor in disease progression and as a potential prognostic marker [51].

One of the unique roles that IL-33 appears to fulfill is its involvement in inducing endocrine resistance, particularly in ER-positive breast cancers [52]. This significant finding is bolstered by observations that cancer relapse in patients undergoing hormone therapy is accompanied by distinctly higher serum IL-33 levels compared to patients who do not experience a relapse [52]. These elevated IL-33 levels seem to be linked to a shortened period of tumor-free survival before a recurrence, indicating the potential detrimental impact of IL-33 on patient prognosis [52]. Supporting these clinical observations, in vitro experiments have further elucidated the role of IL-33 in endocrine resistance. These studies reveal that IL-33 can actively contribute to the induction of endocrine resistance in breast cancer cells, which helps to clarify the correlation between increased IL-33 levels and the incidence of cancer relapse in patients on hormone therapy [52].

Another intriguing aspect of IL-33’s influence is its correlation with resistance to tamoxifen, a common therapeutic challenge in the treatment of ER-positive breast cancers. Furthermore, the overexpression of IL-33 has been associated with an enhancement the in cancer stem cell properties commonly linked to resistance to therapy and disease recurrence [52].

### 3.2. IL-33 and Its Role in Triple-Negative Breast Cancer

Interleukin-33 is becoming increasingly recognized for its crucial role in the dynamics of intratumoral interactions within TNBC, especially in association with the Forkhead Box P3 (FOXP3) protein [53]. The presence of FOXP3+ Tumor-Infiltrating Lymphocytes (TILs) has been associated with the tumor’s expression of IL-33, and intriguingly, the prognostic value of FOXP3 appears to be contingent upon the active expression of IL-33 [53].

IL-33 expression has been observed to be significantly higher in TNBC cell lines in comparison to luminal cell lines. This elevated expression level is correlated with the concentrations of miR-200b, a known positive regulator of IL-33 transcription. However, the levels of miR-200b notably decrease in TNBC cell lines in comparison to non-TNBC cell lines, indicating a potential unique interplay between IL-33 and miR-200b within TNBC [53].

In patient samples acquired prior to Neoadjuvant Chemotherapy (NAC), the presence of both IL-33 and FOXP3 proteins was confirmed, with IL-33 detected primarily in the stromal tumor lesion. Moreover, the TNBC group presented a higher positivity ratio for both FOXP3 and IL-33 when compared to the luminal group, underscoring the potential significance of these proteins in the context of TNBC [53].

In terms of prognostic implications, logistic regression analysis has revealed IL-33 to be a significant predictor of overall survival in patients with TNBC. In particular, patients presenting with high levels of both IL-33 and FOXP3 demonstrated a more favorable prognosis. This suggests that the interaction between IL-33 and FOXP3+ TILs may have substantial implications for therapeutic response and prognosis in TNBC patients [53].

## 4. The Role of the IL-33/ST2 Axis in Breast Cancer Biology 

The role of the IL-33/ST2 axis in breast has been thoroughly studied, and there are much data obtained from both mice models and human breast carcinoma. When it comes to breast carcinoma models, our previous study was the first report to show the time dependent increase in endogenous IL-33 in primary 4T1 mammary tumors in BALB/c mice during cancer progression, as assessed by increased IL-33 mRNA and protein levels accompanied with the time-dependent increase in ST2 mRNA levels [54]. Our investigations have illustrated the IL-33/ST2 axis as a critical promoter of breast cancer proliferation and development by catalyzing the accrual of immunosuppressive cell populations within the tumor microenvironment [54]. Evidence also suggests the influential role of IL-33 in metastasis promotion, with elevated expression of IL-33 observed in metastatic lung niches [54,55]. This cytokine elicits a Th2 immune response, favoring secondary deposit formation [55]. Moreover, within lung metastases, heightened IL-33 activity provokes innate lymphoid 2 cells to secrete IL-13, activating myeloid-derived suppressor cells and sustaining micro- and macro-metastasis [56].

The IL-33/ST2 axis has been implicated in the progression of breast cancer through the induction of tumor tissue inflammation. Such inflammation enhances neoangiogenesis, supports tumor tissue vitality, and reduces tumor necrosis [48]. Another avenue of IL-33 influence is obesity-mediated inflammation in breast carcinoma, inducing the overexpression of IL-33 signaling molecules, which subsequently promote regulatory T cell infiltration, enhancing breast carcinoma aggressiveness [57].

Furthermore, our research indicates the pivotal role of the IL-33/ST2 pathway in promoting mammary tumor growth by upregulating pro-angiogenic VEGF expression in tumor cells and attenuating tumor necrosis [48]. Intriguingly, our results suggest contrasting functions of intracellular and secreted IL-33, hinting at an additional mechanism by which the IL-33/ST2 pathway may be implicated in tumorigenesis [48]. An investigation into a 4T1 breast cancer model revealed significantly higher IL-33 expression within tumor tissue, implying an immune response to 4T1 cells as a potential primary source of IL-33 in the tumor microenvironment [58]. Also, there are data that imply that, both in humans and mice, IL-33/ST2 signaling in breast carcinoma could induce stemness of a malignant cell, making it more challenging to treat with classic therapeutics, such as estrogen receptor inhibitors [58]. 

When it comes to human breast carcinoma, it has been shown that augmented IL-33 expression is associated with endocrine resistance, posing a challenge for some first-line therapeutic modalities [52].

In addition, IL-33/ST2 signaling stimulates the dedifferentiation of malignant breast cancer cells, rendering the disease more aggressive [59]. The overexpression of IL-33 has been observed to activate cancer stem cell genes, such as Aldehyde Dehydrogenase 1 Family Member A3 (ALDH1A3), Octamer-Binding Transcription Factor 4 (OCT4), Nanog Homeobox (NANOG), and SRY-Box Transcription Factor 2 (SOX2), thereby promoting cancer stem cell properties and increasing the difficulty of targeting such cells for treatment [60]. 

Also, elevated IL-33 levels may accelerate breast cancer progression indirectly through signaling molecules like COT (Carcinoma Osaka Thyroid), which stimulate inflammation in the tumor microenvironment and foster the malignant transformation of the mammary epithelium [60].

More recent findings reveal a positive correlation between the overexpression of IL-33 and FOXP3 expression within the tumor microenvironment, suggesting that IL-33 contributes to immunosuppression [53]. Additionally, evidence has emerged indicating that IL-33 might influence breast cancer metabolism, especially by elevating Lipin-1 (LPIN-1) expression, a molecule involved in phospholipid metabolism [61,62]. This alteration helps sustain tumor growth, as LPIN-1, under normal physiological levels, acts as a phosphatase in lipid metabolism and as a transcription factor regulating cell proliferation and differentiation [62]. Notably, the overexpression of LPIN-1 is prevalent in various cancers, particularly breast and prostate carcinoma, and is correlated with poor prognosis [62].

Collectively, these findings illuminate the multifaceted pro-tumorigenic role of IL-33 in breast carcinoma, impacting various aspects of the disease including the tumor tissue, cell proliferation and differentiation, metabolic processes, and immune responses. The numerous functions of IL-33 in the development of breast cancer, as observed in mouse models and human studies, are summarized and detailed in Table 1 and Table 2, respectively. Furthermore, the intricate interactions and influential pathways underpinning the progression of breast carcinoma, with IL-33 at its helm, are schematically represented in Figure 3. By modulating both cancer cells and the anti-tumor immune response, IL-33 emerges as a promising therapeutic target in breast cancer immunotherapy.

### IL-33/ST2 Axis as a Therapeutic Target

Regarding the role of IL-33 in cancer pharmacotherapy, the crux is understanding its direct and indirect effects. Anti-IL-33 and anti-ST2 antibodies are examined in the therapy of various inflammatory diseases, such as asthma, chronic obstructive pulmonary disease (COPD), atopic dermatitis, etc., in the second phase of clinical studies; however, their role in cancer therapy has not been fully investigated [10]. In several preclinical in vitro and in vivo models of melanoma [63,64], colon [65], and non-small-cell lung cancer [66], tumor suppression was observed after treatment with the aforementioned antibodies [63]. Promising antitumor achievements based on the blockade of the IL-33/ST2 axis were also determinate in breast cancer [67]. The antitumor effects of the IL-33/ST2 axis are discussed as well and linked to the activation of immune effector cells [63,64,68,69,70]. The individual results indicate that the administration of IL-33 could suppress the development of breast tumor metastases in mice lungs [71]. On the other hand, some evidences suggest that the application of IL-33 promotes metastasis and breast cancer growth [54,61]. The role of fibroblast-derived IL-33 in potentiating the occurrence of breast cancer metastases in the lungs was pointed out [55]. Bearing in mind that high levels of IL-33 may be associated with serious inflammatory processes, and the aforementioned conflicting findings, the possibility of using IL-33 in tumor immunotherapy still needs to be addressed [19]. IL-33 has been linked to the resistance of melanoma and breast cancer cells to conventional pharmacotherapeutics [63], which might indicate the possibility of targeting the IL-33/ST2 axis as an immune antitumor adjuvant. Kudo-Saito, C et al. reported that soluble IL-33 released from cancer cells stimulates the expression of ST-2 on IL17 Receptor B positive (IL17RB+) and GATA Binding Protein 3 positive (GATA3+) cells, thus leading to tumor progression [63]. A recent study suggested that cancer-cell-derived IL-33 is necessary to synergize with Cytotoxic T-Lymphocyte-Associated protein 4 (CTLA-4) or Programmed Cell Death 1 (PD-1) monoclonal antibodies (mAbs) to enhance antitumor efficacy [72]. Also, IL-33 has been recognized as a predictive marker for sensitivity to 5-fluorouracil via T cell responses in colorectal cancer [73]. In triple-negative breast cancer, a high expression of IL-33 in tumor tissue was a predictor of the response to chemotherapy [53]. In breast cancer cells isolated from patients, IL-33 levels could be linked to future tamoxifen resistance [52]. Although the mechanism has not been fully elucidated, it is considered that IL-33 could induce specific properties of breast cancer stem cells [52]. Hollande, C et al. reported that the administration of a novel antidiabetic drug, sitagliptin, was associated with enlarged eosinophils migration into solid tumors and that IL-33 was necessary for anti-tumor responses mediated by eosinophils [12]. Eosinophil-maintained breast cancer immunotherapy was uncovered by Blomberg et al. [74]. Eosinophils might enhance the effects of immune checkpoint inhibitors upon IL-33 and IL-5 stimulation [74,75]. In triple-negative breast cancer patients and in vivo mouse models, IL-33 engages eosinophil infiltration into tumors and is essential for the efficiency of cisplatin and anti-PD1 and anti-CTLA4 treatment [74]. Moreover, targeting the IL33-ST2 axis with the additional use of checkpoint inhibitors has been confirmed to be a promising strategy for tumor immunotherapy [76]. The dual blockage of the PD-L/PD-1 and IL33/ST2 axes inhibited the progression of mice breast cancer [76]. 

The complex and multifaceted roles of IL-33 in cancer pharmacotherapy necessitate a thorough understanding and careful consideration of its direct and indirect effects. The IL-33/ST2 axis represents a dynamic and promising target in the field of cancer therapy, especially in the context of breast cancer. Conflicting findings and the association of high levels of IL-33 with serious inflammatory processes indicate that the utility of targeting IL-33 in tumor immunotherapy remains a subject of intense investigation. The potential therapeutic interventions directed towards IL-33 are summarized and detailed in Table 3. The insights gained from recent studies provide an essential foundation for future research in immuno-oncology, highlighting the potential of the IL-33/ST2 axis as an innovative therapeutic avenue.

## 5. Future Perspectives in the IL-33/ST2 Axis and Breast Carcinoma Research

The intricate involvement of the IL-33/ST2 axis in breast carcinoma has begun to unravel a plethora of research avenues and therapeutic strategies. This study has highlighted the prospective areas of investigation and their potential implications:Elucidating Mechanisms: Our current comprehension of the IL-33/ST2 axis in breast carcinoma, while extensive, is yet incomplete. It is imperative that future research dives deeper into the molecular pathways influenced by the IL-33/ST2 axis, particularly those that might govern therapeutic resistance or susceptibility.Therapeutic Targeting: The centrality of the IL-33/ST2 axis in shaping the immune response in breast cancer lends itself as an attractive target for therapy. Potential interventions could range from direct IL-33 inhibitors and neutralizing antibodies to antagonists that block ST2-mediated signaling, thereby neutralizing its pro-tumorigenic activities.Biomarker Potential: Given the differential expression of IL-33 and ST2 across breast carcinoma subtypes, there is potential to employ the IL-33/ST2 axis as a diagnostic or prognostic biomarker. Multi-center studies assessing IL-33 and ST2 levels could shed light on their role in disease trajectory, therapeutic responsiveness, or potential recurrence.Synergy with Current Therapies: The prospect of combining therapies targeting the IL-33/ST2 axis with existing immunotherapies holds promise. For instance, integrating IL-33 inhibition with immune checkpoint blockade might unveil unforeseen synergistic effects in tumor suppression.Role in Metastasis: The IL-33/ST2 axis’ potential role in driving metastatic tendencies, especially to organs such as the lungs, necessitates comprehensive exploration. Understanding how this axis participates in establishing metastatic niches could inform therapeutic strategies to counteract these processes.Tumor Microenvironment Modulation: The dynamic interplay between the IL-33/ST2 axis, the TME, and its constituent immune cells (like tumor-infiltrating lymphocytes, macrophages, and dendritic cells) needs detailed investigation. Understanding how the IL-33/ST2 axis modulates the TME can provide insights into strategies to render the TME unfavorable for tumor advancement.Patient Stratification: Given breast cancer’s heterogeneity, determining treatment strategies based on IL-33 and ST2 expression, or the resulting downstream effects, could lead to more personalized and effective therapeutic regimens.

Moving forward, the challenge lies not just in understanding the role of the IL-33/ST2 axis in breast carcinoma, but in translating this knowledge into tangible clinical benefits. It is our hope that a collaborative, interdisciplinary approach will expedite the realization of this potential, leading to more targeted and effective therapeutic interventions in the near future

## 6. Conclusions

Our understanding of the IL-33/ST2 axis in breast cancer has come a long way, as we are uncovering its multifaceted roles in tumor growth, metastasis, and immunosuppression. The complex nature of this axis suggests that it could serve as a potential prognostic marker of, as well as a therapeutic target in, breast cancer. Despite the encouraging results observed in preclinical models, much remains to be elucidated regarding the diverse roles of IL-33/ST2 in different cancer stages, types, and responses to treatment. The conflicting roles of IL-33 also raise crucial questions about the best strategies for its clinical application. Future research should aim to elucidate the intricate balance of pro-tumorigenic and anti-tumorigenic effects mediated by the IL-33/ST2 axis and to focus on how to harness this knowledge to improve the management of breast cancer. Furthermore, clinical trials are required to validate these findings in human patients and to determine the safety and efficacy of potential therapeutic strategies targeting this axis.

## Figures and Tables

**Figure 1 ijms-24-14026-f001:**
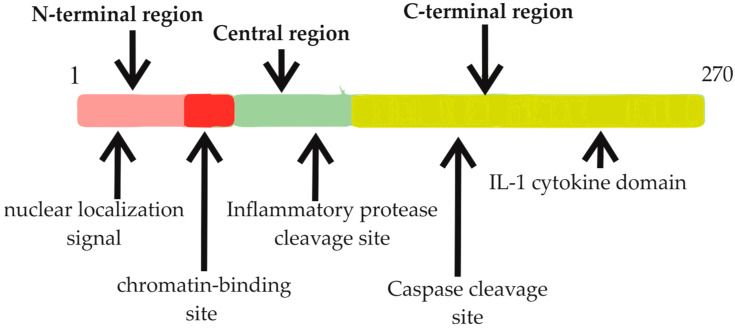
Primary structure of the human IL-33 protein. The protein structure is delineated into three principal domains: the N-terminal domain, which encompasses the nuclear localization signal and the chromatin binding site; the central domain, highlighting the cleavage sites for inflammatory proteases involved in IL-33 activation; and the C-terminal domain, characterized as the IL-1-like cytokine domain, with noted cleavage sites for caspases responsible for IL-33 inactivation.

**Figure 2 ijms-24-14026-f002:**
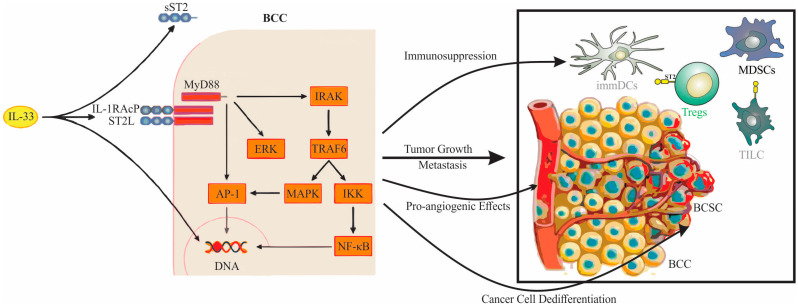
IL-33/ST2 signaling cascade and associated outcomes in breast cancer. On the left side of the figure, the signaling cascade following the binding of IL-33 to the receptor complex is illustrated. Upon receptor activation, the intracellular signaling components, including MyD88, IRAK, ERK, and TRAF6, sequentially lead to the activation of NF-κB and MAPK pathways. On the right side of the figure, the outcomes of this signaling are depicted, including tumor growth and metastasis, immunosuppression, pro-angiogenic effects, and cancer cell dedifferentiation. Note: IL-33 (Interleukin-33), ST2L (Transmembrane ST2 receptor), sST2 (Soluble ST2), IL1RaCP (Interleukin-1 Receptor Accessory Protein), BCC (Breast Cancer Cell), BCSC (Breast Cancer Stem Cell), MyD88 (Myeloid Differentiation Primary Response 88), IRAK (Interleukin-1 Receptor-Associated Kinase), ERK (Extracellular Signal-Regulated Kinase), TRAF6 (TNF Receptor Associated Factor 6), AP-1 (Activator Protein 1), MAPK (Mitogen-Activated Protein Kinase), IKK (IκB Kinase), NF-κB (Nuclear Factor kappa-light-chain-enhancer of activated B cells), DNA (Deoxyribonucleic Acid), MDSCs (Myeloid-derived Suppressor Cells), Tregs (Regulatory T Cells), immDCs (Immature Dendritic Cells), and TILC (Tumor-infiltrating Innate Lymphoid Cells). The figure offers a comprehensive view of the IL-33/ST2 axis and how it contributes to various aspects of breast cancer pathogenesis.

**Figure 3 ijms-24-14026-f003:**
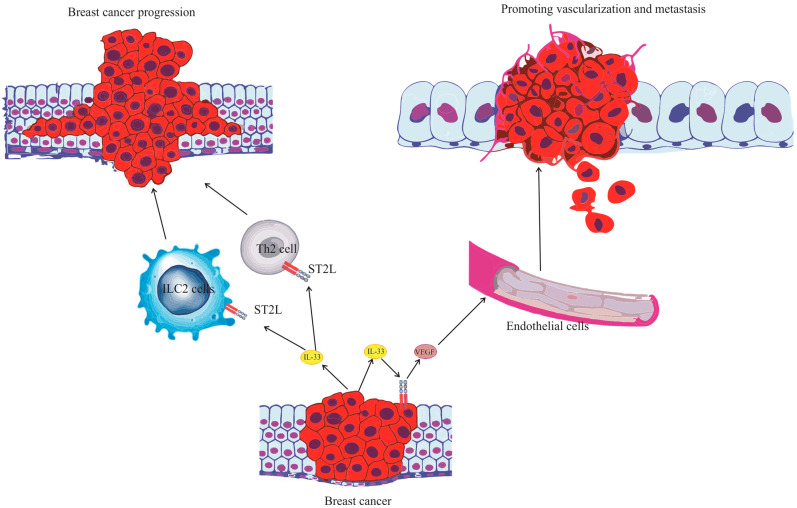
Interplay of IL-33/ST2 Axis in breast carcinoma progression. Schematic representation illustrating the intricate interactions underpinning breast carcinoma progression. The central portrayal of an anomalous breast cancer cell emphasizes its principal interactions with surrounding molecules and cells. IL-33, emanating from the tumor, serves a pivotal role in modulating the tumor microenvironment and promotes the accumulation of immunosuppressive cells. Specifically, cells like Innate Lymphoid Cells 2 (ILC2) and T helper 2 (Th2) are attracted to the environment, underscoring their potential role in dampening the immune response against the tumor, which further bolsters tumor growth progression. In tandem, IL-33 is linked with a heightened expression of Vascular Endothelial Growth Factor (VEGF) molecules from the cancer cell, hinting at its role in fostering angiogenesis and boosting metastatic potential. The figure accentuates the multifaceted interplay and determinants shaping the progression of breast cancer.

**Table 1 ijms-24-14026-t001:** Role of IL-33 in breast carcinoma development: insights from mouse models.

Role of IL-33	Mechanism	Reference
Promotion of breast cancer proliferation	Stimulates the accumulation of immunosuppressive cell populations in the tumor microenvironment	[54]
Promotion of metastasis	Th2 immune response favoring secondary deposit formation, elicits IL-13 from innate lymphoid 2 cells	[54,55,56]
Induction of tumor tissue inflammation	Enhances neoangiogenesis, supports tumor vitality, reduces tumor necrosis	[48]
Promotion of mammary tumor growth	Upregulates VEGF expression in tumor cells, attenuates tumor necrosis	[48]
Induction of malignant cell stemness	Induces stemness, making treatment with classic therapeutics, such as estrogen receptor inhibitors, more challenging	[58]

**Table 2 ijms-24-14026-t002:** Impact of IL-33 on breast carcinoma progression: observations from human studies.

Role of IL-33	Mechanism	Reference
Association with endocrine resistance	Challenges some first-line therapeutic modalities	[52]
Dedifferentiation of malignant breast cancer cells	Activates cancer stem cell genes like ALDH1A3, OCT4, NANOG, and SOX2	[59,60]
Acceleration of breast cancer progression	Signals molecules like COT to stimulate inflammation in the tumor microenvironment	[60]
Contribution to immunosuppression	Positive correlation with FOXP3 expression within the tumor microenvironment	[53]
Influence on breast cancer metabolism	Elevation of LPIN-1 expression, a molecule involved in phospholipid metabolism	[61,62]

**Table 3 ijms-24-14026-t003:** Potential therapeutic interventions targeting IL-33.

Therapeutic Intervention	Mechanism
Blocking IL-33/ST2 Signaling	Inhibits the dedifferentiation of malignant breast cancer cells
Inhibition of IL-33 Expression	Targets cancer stem cells by reducing the activation of key stem cell genes
Regulation of IL-33 Induced Signaling Molecules	Decreases inflammation in the tumor microenvironment and prevents malignant transformation

## Data Availability

This review article is based on previously conducted studies and does not contain any original data collected by the authors. All information and data analyzed in this study are cited and available in the referenced literature.

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
