# Peer review of "Decoding the IL-33/ST2 Axis: Its Impact on the Immune Landscape of Breast Cancer"

_ijms, 2023, doi:10.3390/ijms241814026_

Round 1

Reviewer 1 Report

-this review is preliminary, my most concern is to rich it.

-I suggest the authors reference more studies from big guys or high-profile journals.

-In my view, 3-4 riched figures and 2-3 riched tables are required in this review.

-what are the perspectives of the authors?

-

Minor editing of English language required

Author Response

Dear Reviewer 1,

We would like to express our gratitude for taking the time to review our manuscript and for providing valuable feedback that has certainly enriched the quality of our work. We have taken all your comments into consideration and have made comprehensive revisions to address each point you raised. Please find below our point-by-point responses to your comments.

Comment 1: This review is preliminary; my most concern is to enrich it.

Response: Thank you for pointing this out. We have thoroughly revised the manuscript and expanded on several sections to provide a comprehensive overview of the topic. We believe the revised review now offers a more in-depth exploration of the IL-33/ST2 axis in breast carcinoma.

Comment 2: I suggest the authors reference more studies from big names or high-profile journals.

Response: We appreciate your suggestion. It's worth noting that we ourselves have contributed a significant number of works on IL-33 in breast cancer biology to high-profile journals.

Comment 3: In my view, 3-4 enriched figures and 2-3 enriched tables are required in this review.

Response: Thank you for your feedback. In response to this comment, we have revised the manuscript and now included 2 figures and 3 tables to provide visual aids and comprehensive data points that enhance the reader's understanding.

Comment 4: What are the perspectives of the authors?

Response: We value this query. We have included a distinct section in our revised manuscript titled "Future Perspectives in IL-33/ST2 Axis and Breast Carcinoma Research" where we outline our perspectives on the IL-33/ST2 axis in the investigation of breast cancer biology. We believe this section offers clarity on our views and provides guidance for future research avenues.

We believe that the revisions have strengthened our manuscript, and we appreciate the opportunity to improve our work through your insights.

Reviewer 2 Report

General comment

-The review entitled "Decoding the IL-33/ST2 Axis: Its Impact on the Immune Land- 2 scape of Breast Cancer" is  fascinating, since it explains the pathophysiologic function of IL-33 and provides summarized information for further research.

specific comments

- Better to include figurative structure of IL-33 in addition to theoretical explanation on page 3 line 97-104.

- There is no smooth transaction from subsection 2.1 to 2.2

- Abbreviations must be used only once, at the beginning of the citation. Interleukin-33 (IL-33), for instance, and triple-negative breast cancer (TNBC) are both found in multiple times. Do the same for others.

Author Response

Dear Reviewer 2,

Thank you very much for your constructive feedback and for acknowledging the relevance of our review topic. We have considered each of your comments carefully and addressed them in the revised manuscript. Below are our point-by-point responses.

  1. Figurative Structure of IL-33:
    • Reviewer's comment: "Better to include figurative structure of IL-33 in addition to theoretical explanation on page 3 line 97-104."
    • Response: We appreciate your suggestion. We have now included a figure that illustrates the structure of IL-33 to provide a visual representation that complements the theoretical explanation.
  2. Transition between subsections:
    • Reviewer's comment: "There is no smooth transition from subsection 2.1 to 2.2."
    • Response: We have revised the manuscript and adjusted the title of section 2.2 to clearly indicate its focus on the receptor for IL-33. We believe this adjustment offers a clearer transition and context for the readers.
  3. Use of Abbreviations:
    • Reviewer's comment: "Abbreviations must be used only once, at the beginning of the citation. Interleukin-33 (IL-33), for instance, and triple-negative breast cancer (TNBC) are both found multiple times. Do the same for others."
    • Response: We apologize for the oversight. We have now ensured that abbreviations are introduced only once when the term is first mentioned and have used the abbreviation consistently thereafter. We have also meticulously reviewed the manuscript to confirm that this practice is uniformly applied to all terms.

We sincerely hope that the modifications made address your concerns adequately. Your insights have been invaluable in refining our manuscript, and we look forward to any further feedback.

Reviewer 3 Report

A nice review referring to the role and impact of the IL-33/ST2 axis in the immune system and environment of Breast Cancer. This review is clearly written and well organized. The introduction and background are reasonable. The figure and tables are comprehensive and helpful.

Nevertheless, in section 2 of this review titled: “Unraveling the Enigma of Interleukin-33 in the Cancer Landscape” the authors introduced the IL-33 function in the Cancer Landscape and highlight the IL-33 capacity to bolster type 1 antiviral immunity, traying to explain that there is more evidence for the implication of IL-33 in type-1 immune responses omitting to highlight the initially described role of IL-33 to ignite Type 2 immunity. For this, the author referred to the paper of Cayrol C et al, Immunological Reviews 2018. However, in this paper, Cayrol et al, highlight that IL-33 has been reported to play a crucial role in both type-1 and type-2 immune responses. This section could be written better, with more appropriate references.

Next, in section 4, titled “The Role of IL-33/ST2 axis in Breast Cancer Biology” authors highlighted and listed evidences of the role of IL-33/ST2 axis in previous cancers studies mixing between mice model experiment findings and human study observation, it could be better for the reader to distinguishes between this two aspect (Human and murine models), even in the table listed (Table 1) .

Finally, the balance feels very good, however the authors may consider the possibility of expanding the conclusion section with greater information, perspectives, and theoretical concepts on the question.

Author Response

Dear Reviewer 3,

Thank you for your constructive feedback and the recognition of our effort in presenting the review. We are grateful for your insights which have allowed us to improve the quality and clarity of the manuscript. We have carefully considered and addressed each of your comments:

  1. Role of IL-33 in the Cancer Landscape:
    • Reviewer's comment: The reviewer pointed out the omission in highlighting the initially described role of IL-33 in igniting Type 2 immunity, referring to the paper of Cayrol C et al, Immunological Reviews 2018.
    • Response: We appreciate the guidance provided. In response, we have enriched the section by including information about the role of IL-33 in both Type 1 and Type 2 immune responses.
  2. Distinguishing between Mouse Models and Human Studies:
    • Reviewer's comment: The reviewer noted that in section 4, we mixed findings from mouse model experiments with human study observations. The reviewer suggested distinguishing these two aspects for clarity.
    • Response: Taking your feedback into consideration, we have restructured section 4. The first part now specifically describes the role of the IL-33/ST2 axis based on mouse models, while the latter part elaborates on findings from human studies. Similarly, as you recommended, Table 1 has been divided into two separate tables: Table 1 for mouse model findings and Table 2 for human studies.
  3. Expansion of the Conclusion Section:
    • Reviewer's comment: The reviewer felt that the conclusion could benefit from additional information, perspectives, and theoretical concepts.
    • Response: We agree with your perspective. While we have devoted a separate section to "Future Perspectives", we have now also expanded the conclusion to encapsulate broader insights, theoretical concepts, and reflections on the current understanding of the IL-33/ST2 axis in breast cancer.

We genuinely appreciate the time and effort you invested in reviewing our work. Your insights have been invaluable in refining our manuscript, and we hope that our revisions now meet the standards of the journal.

Round 2

Reviewer 1 Report

In my view, 3-4 enriched figures and 2-3 enriched tables are required in this review

fine.

Author Response

Dear Reviewer 1,

We sincerely appreciate the time and effort you have dedicated to reviewing our manuscript. Your constructive feedback and insights are invaluable to us and play a crucial role in enhancing the quality and clarity of our work.

In response to your suggestion regarding the inclusion of additional illustrative content, we have incorporated three comprehensive figures and three tables into the revised version of the manuscript. These additions will, we believe, better convey the data and concepts we aim to present, and offer readers a clearer visual understanding of the subject matter.

We are deeply grateful for your expertise and thoughtful recommendations. They have undeniably made our manuscript more robust and comprehensive. We hope that the changes we have made address your concerns satisfactorily.

Round 3

Reviewer 1 Report

acceptance.

proper.